# Vanadium Extraction from the Flotation Concentrate of Vanadium-Bearing Shale by Process of Non-Roasting Enhanced Acid Leaching and Thermodynamics

**DOI:** 10.3390/molecules28062706

**Published:** 2023-03-16

**Authors:** Yue Tang, Guohua Ye, Qi Zuo

**Affiliations:** Faculty of Land Resource Engineering, Kunming University of Science and Technology, Kunming 650093, China

**Keywords:** vanadium-bearing shale, flotation concentrate, garnet, enhanced acid leaching, thermodynamics

## Abstract

The purpose of this work is to significantly improve the vanadium grade of vanadium-bearing shale after flotation preconcentration, which is conducive to reducing the acid consumption and industrial costs in the subsequently enhanced acid leaching of vanadium. Vanadium concentrate from vanadium-bearing shale enriched by flotation is used for acid-leaching feed. The leaching effects of two kinds of acid-leaching systems were compared, and the mechanism of acid leaching on the mineral structure was also described. The difficulty of spontaneous reactions of vanadium-bearing minerals such as garnet in an acid-leaching system was studied through thermodynamics. Additionally, several characterization methods were used to evaluate the improvement of leaching performance. The addition of oxidants and fluorinating aids strengthens the acid-leaching process, which greatly destroys the structure of a garnet, which is conducive to the extraction of vanadium in a flotation concentrate. The leaching efficiency can reach 94.86%, and the acid consumption is also reduced. Through the mechanism study of the leaching system, it is expected that when the enhanced acid-leaching process is put into industrial production, the effective leaching of vanadium can be accurately controlled, and the difficulty of subsequent vanadium enrichment and purification can be reduced.

## 1. Introduction

Vanadium is a rare strategic metal with multivalent states. China, the United States, Japan, and the European Union have listed it in the directories of strategic metals to safeguard national security [1]. With the rapid evolution of global strategic emerging industries, the development and utilization of vanadium resources have attracted much attention [2]. In China, the reserves of vanadium-bearing shale are up to 61.88 billion tons. Vanadium-bearing shale is a unique strategic advantageous resource to produce vanadium products. Recovering and extracting the valuable metal of vanadium from vanadium-bearing shale has significant economic benefits and social value [3,4,5].

At present, roasting–leaching is commonly used to extract vanadium from vanadium-bearing shale. The non-salt roasting–leaching method is simple, with less pollution to the environment, but the conversion leaching rate of vanadium is low [6,7]. In contrast, salt roasting–leaching can effectively destroy the crystal structure of vanadium-bearing minerals [8,9,10]. However, for salt roasting–leaching, a large number of impurity metal ions infiltrate into the leaching solution, which is not conducive to the purification and enrichment of the leaching solution. To solve the problems caused by roasting–leaching, it is necessary to develop a green, clean, efficient, and energy-saving extraction process for vanadium-bearing shale [11]. Extracting vanadium by acid leaching without roasting can save the equipment for roasting minerals and avoid environmental pollution, which has become an important development direction of vanadium extraction. Pressure, microwave, ultrasonic, and other auxiliary methods in the leaching process are often used to obtain a sufficient leaching efficiency of vanadium [12,13,14,15]. However, these methods not only have precision for equipment and fine requirements operation but also cause high-producing costs. In acid leaching with leaching aids, the process can extremely improve the leaching efficiency of vanadium, the cost being a lower and simple operation. Therefore, the method of adding leaching aids in the acid-leaching process is widely used [16].

Oxidizing aids and fluorine-bearing leaching aids are two types of leaching aids widely used in the hydrometallurgical industry [17,18]. Dai et al. added mixed oxidant to acid-leaching vanadium from a stone coal vanadium ore in southwest China to improve the vanadium leaching rate, and Li et al. used sodium fluoride as a leaching agent for acid-leaching a low-grade vanadium-bearing shale. Under optimal leaching conditions, the leaching efficiency of vanadium is up to 92.96%.

As we all know, the commonly used oxidizing aids are MnO_2_ and NaClO_3_, which can oxidize most of the low-valence vanadium into its high-valence form and improve the leaching efficiency. In an acid-leaching system, MnO_2_ shows stronger oxidation, lower toxicity, and a stable state.

Fluorine used in China’s industrial production is mainly supplied by fluorite (CaF_2_), which is rich in resource reserves and widely distributed. Therefore, in the acid-leaching process, the leaching aids of MnO_2_ and CaF_2_ were selected in experiments to enhance the ability to destroy the crystal structure of vanadium-bearing minerals. Consequently, vanadium in the minerals could be efficiently released, thus improving the leaching efficiency of vanadium. Finally, the effects of the dosage of sulfuric acid, leaching time, liquid–solid ratio, agitation speed, temperature, and other factors on the leaching efficiency of vanadium for the flotation concentrate of vanadium-bearing shale in the absence and presence of leaching aids were studied through leaching experiments. Through X-ray diffraction (XRD), atomic force microscopy (AFM), Fourier transform infrared spectroscopy (FT-IR), scanning electron microscopy (SEM), and other detection methods, as well as the calculation of related thermodynamic parameters, the enhancing mechanism of acid-leaching aids for the flotation concentrate of vanadium bearing shale was revealed.

## 2. Results and Discussion

### 2.1. Thermodynamics

#### 2.1.1. Basic Analysis of Gibbs Free Energy Change

Whether a chemical reaction can proceed spontaneously can be determined by the change in the standard Gibbs free energy ($\Delta G_{T}^{\theta }$), the formula for which is as follows [19]:(1)$$\Delta G_{T}^{\theta }=\Delta H_{T}^{\theta }-T\Delta S_{T}^{\theta }$$

In this equation, when $\Delta G_{T}^{\theta }$ > 0, the reaction cannot proceed spontaneously. When $\Delta G_{T}^{\theta }\;$= 0, the reaction can maintain a balanced state. When $\Delta G_{T}^{\theta }$ < 0, the reaction can be spontaneous.

In Equation (1), $\Delta H_{T}^{\theta }$ and $\Delta S_{T}^{\theta }$ represent the enthalpy change (kJ/mol) and entropy change (kJ/mol) of the chemical reaction, respectively, at a reaction temperature T. The calculation formulas of $\Delta H_{T}^{\theta }$ and $\Delta S_{T}^{\theta }$ are shown in Equations (2) and (3).
(2)$$\Delta H_{T}^{\theta }=\Delta H_{298}^{\theta }+\int_{298}^{T}\Delta C_{P}\mathrm{dT}$$
(3)$$\Delta S_{T}^{\theta }=\Delta S_{298}^{\theta }+\int_{298}^{T}\frac{\Delta C_{p}}{T}\mathrm{dT}$$

Equation (4) is obtained by summing Equations (1)–(3):(4)$$\Delta G_{T}^{\theta }=(\Delta H_{298}^{\theta }+\int_{298}^{T}\Delta C_{p}\mathrm{dT})-T(\Delta S_{298}^{\theta }+\int_{298}^{T}\frac{\Delta C_{p}}{T}\mathrm{dT})$$

In this study, $\Delta C_{p}$ represents the specific heat tolerance. Finally, the relationship between ∆G and T is calculated by *HSC Chemistry 9.0* developed by *Finland Outokumpu Company* to explore the effect of temperature change on the leaching performance.

#### 2.1.2. Dissolution Reaction of Vanadium-Bearing Mica

According to the results of phase analysis in Table 1, vanadium-bearing mica (K(V, Al, Mg)_2_AlSi_3_O10(OH)_2_)muscovite and biotite) are the main vanadium-bearing minerals. Hence, the process of vanadium dissolution is closely related to the dissolution reaction of vanadium-bearing mica. Among them, the dissolution reaction of vanadium-bearing mica mainly includes four metals: K, V, Al, and Mg. However, Mg and K are easily soluble in water, and their reactions in an aqueous solution are relatively simple, so they will not be discussed. Therefore, this section focuses on the dissolution reaction of V and Al.

##### Dissolution Reaction of Vanadium

Under different pH conditions, the existing form of vanadium in an aqueous solution shows variability, so its electrochemical reaction is more complex. A series of the main electrochemical reaction equations in an aqueous solution are shown in Table 1 [20]. Under the conditions of 25 °C and 1 bar, the electric potential (Eh)-pH diagram of the V-H_2_O system was drawn according to the main equations listed in Table 1 combined with *HSC Chemistry 9.0 software*. The detailed calculation results are shown in Figure 1.

Figure 1 mainly shows the relationship between Eh and pH of the V-H_2_O [21] system. It can be seen that when the aqueous solution is in a strong acid state (pH < 1), vanadium will be converted into cationic forms of different valence states with the change in the Eh from low to high. Among them, V (II) is stable below the H_2_ formation in the form of V^2+^ in all pH ranges and can be displaced H^+^ from the aqueous solution. Thus, it has strong reducibility. V^3+^ is easily oxidized in an alkaline system and can only exist in a strong acid system. However, with the decrease in pH, it will be hydrolyzed into VOH^2+^ and further decomposed into VO^+^. When −2 < pH < 4, V (IV) exists in a large range and stably in the aqueous solution as VO^2+^. When 4 < pH < 14, vanadium will be hydrolyzed into a dimer (HV_2_O_5_^−^).

The charge radius of V (V) is large, so it is easy to hydrolyze. Therefore, during the transition from an acidic system to an alkaline system, high-valence V (V) occurs in various forms in the aqueous solution. However, it mainly exists in the form of oxygen binding to form vanadium oxide ions or vanadate ions, and high-valence VO_2_^+^ only appears in the high-acid (pH < 2) and high-potential areas. Therefore, in order to facilitate the subsequent enrichment and separation process, it is essential to adjust Eh and pH values in the leaching solution to obtain high-valence vanadium. (Setting the total content of vanadium as 1 mol.)

##### Dissolution Reaction of Aluminum

The electrochemical reaction equation of aluminum in an aqueous solution is shown in Table 2. Under the conditions of 25 °C and 1 bar, the Eh-pH diagram of the Al-H_2_O system was drawn according to the main equations listed in Table 2 and *HSC Chemistry 9.0 software*. The detailed calculation results are shown in Figure 2.

Figure 2 shows the Eh–pH relationship of the Al-H_2_O system. Al exists in an environment with pH < 8, and its Eh is considerably lower than that of H_2_. Although Al^3+^ exists in a strong acid environment with pH < 4, in a strong alkali system, it exists in the form of Al(OH)_4_^-^ in an aqueous solution. Meanwhile, Al(OH)_3_ can exist under both acidic and alkaline conditions with a large advantageous area. Therefore, if Al exists in the leaching solution in the form of Al^3+^, it must be removed, considering the difficulty involved in the purification and enrichment of the obtained leaching solution. Therefore, by adjusting the pH value to be greater than 2.6, Al will be released in the form of Al(OH)_3_ precipitation so that it will not enter the leaching solution with vanadium.

##### Gibbs Free Energy of Vanadium-Bearing Mica Dissolution Reaction

Herein, muscovite as a representative of vanadium-bearing mica was considered the research object. Under high temperature and strong acid, the performance of the two leaching systems with and without leaching aids was compared. Furthermore, the effect of the leaching aids addition on the dissolution reaction of white mica was explored, and the Gibbs free energy of the muscovite dissolution reaction was investigated to understand the mechanism of vanadium-bearing mica in strengthening acid leaching. Equations (29)−(43) refers to the dissolution reaction of two acid leaching systems of muscovite without adding fluorine-bearing leaching aid and adding fluorine-bearing leaching aid, respectively. (M and a represent solid state and aqueous solution, respectively).
KAl2(AlSi3O10)(OH)2(M) + 10H(+a) = K(+a) + 3Al(+3a) + 3H4SiO4(a)(29)
KAl2(AlSi3O10)(OH)2(M) + 12HF(a) + 10H(+a) = K(+a) + 3Al(+3a) + 3SiF4(g) + 12H2O(l)(30)
KAl2(AlSi3O10)(OH)2(M) + 15HF(a) + 7H(+a) = K(+a) + 3AlF(+2a) + 3SiF4(g) + 12H2O(l)(31)
KAl2(AlSi3O10)(OH)2(M) + 18HF(a) + 4H(+a) = K(+a) + 3AlF2(+a) + 3SiF4(g) + 12H2O(l)(32)
KAl2(AlSi3O10)(OH)2(M) + 21HF(a) + H(+a) = K(+a) + 3AlF3(a) + 3SiF4(g) + 12H2O(l)(33)
KAl2(AlSi3O10)(OH)2(M) + 22HF(a) + 2F(−a) = K(+a) + 3AlF4(−a) + 3SiF4(g) + 12H2O(l)(34)
KAl2(AlSi3O10)(OH)2(M) + 22HF(a) + 5F(−a) = K(+a) + 3AlF5(−2a) + 3SiF4(g) + 12H2O(l)(35)
KAl2(AlSi3O10)(OH)2(M) + 22HF(a) + 8F(−a) = K(+a) + 3SiF4(g) + 3AlF6(−3a) + 12H2O(l)(36)
KAl2(AlSi3O10)(OH)2(M) + 18HF(a) + 4H(+a) = K(+a) + 3Al(+3a) + 3SiF6(−2a) + 12H2O(l)(37)
KAl2(AlSi3O10)(OH)2(M) + 21HF(a) + H(+a) = K(+a) + 3AlF(+2a) + 3SiF6(−2a)12H2O(l)(38)
KAl2(AlSi3O10)(OH)2(M) + 22HF(a) + 2F(−a) = K(+a) + 3AlF2(+a) + 3SiF6(−2a) + 12H2O(l)(39)
KAl2(AlSi3O10)(OH)2(M) + 22HF(a) + 5F(−a) = K(+a) + 3AlF3(a) + 3SiF6(−2a) + 12H2O(l)(40)
KAl2(AlSi3O10)(OH)2(M) + 22HF(a) + 8F(−a) = K(+a) + 3AlF4(−a) + 3SiF6(−2a) + 12H2O(l)(41)
KAl2(AlSi3O10)(OH)2(M) + 22HF(a) + 11F(−a) = K(+a) + 3AlF5(−2a) + 3SiF6(−2a) + 12H2O(l)(42)
KAl2(AlSi3O10)(OH)2(M) + 22HF(a) + 14F(−a) = K(+a) + 3AlF6(−3a) + 3SiF6(−2a) + 12H2O(l)(43)

According to Equations (29)–(43), the relationship between ∆G and T (temperature) was plotted using the *HSC Chemistry 9.0* software Reaction Equations module, as shown in Figure 3.

It can be seen from Figure 3a,b that when the temperature increases from 55 °C to 100 °C, ∆G with and without a fluorine-bearing leaching aid, all are decreased, and both values are less than 0. This shows that the dissolution reaction of two acid-leaching systems of mica is spontaneous, and the ∆G value with fluoride-bearing leaching aids is significantly lower than that without its addition. When the temperature is 95 °C, ∆G of Equation (29) is −73.497 kJ/mol, ∆G of Equations (30)–(36) decreases from −104.385 kJ/mol to −430.957 kJ/mol, and ∆G of Equations (37)–(43) decreases from −107.755 kJ/mol to −249.787 kJ/mol. With the addition of F^-^, ∆G of the reaction equation decreases, and the minimum energy drops to −430.957 kJ/mol, which is more than 5 times higher than the ∆G of Equation (29). This is because the addition of F^−^ will generate HF to adsorb on the mica surface, and Si on the mica surface will be corroded by HF, so the Si–O bond will break. Moreover, F^−^ having strong electronegativity can accelerate the fracture of the Al–O and Si–O bonds in the crystal structure of muscovite. Therefore, F^−^ can cause serious damage to the crystal lattice of muscovite, and its dissolution reaction is carried out more easily in a positive direction so that vanadium is dissolved effectively.

#### 2.1.3. Dissolution Reaction of Vanadium-Bearing Garnet

Garnet (Ca_3_(Al,Cr,V)_2_[SiO_4_]_3_) can be decomposed into the form of a metal oxide [3CaO·2(Al_2_O_3_,Cr_2_O_3_,V_2_O_3_)·3SiO_2_]. It belongs to island silicate minerals that are difficult to leach. Because vanadium in island aluminosilicate minerals exists in the form of a homogeneous phase replacement and is more stable, the crystal structure of a garnet is difficult to destroy. A total of 21.1% of vanadium (Table 3) in original samples is distributed in garnet. Consequently, leaching aids were added to strengthen the acid leaching process to help release vanadium. Therefore, the acid-leaching process intensification was verified by calculating the relevant thermodynamic parameters of garnet. The dissolution reaction of Al and V has been described in detail in the previous section, so it is not described here. In addition, the relevant electrochemical reaction of Ca in an aqueous solution is relatively simple and is not explained here. This section mainly describes the dissolution reaction of Cr.

##### Dissolution Reaction of Chromium

The electrochemical reaction equation of chromium in an aqueous solution is shown in Table 3. Under the conditions of 25 °C and 1 bar, the Eh-pH diagram of the Cr-H_2_O system was obtained according to the main equations listed in Table 3 and *HSC Chemistry 9.0* software. The detailed calculation results are shown in Figure 4.

It can be seen from Figure 4 that Cr in the Cr-H_2_O [22,23] system is in the immunity area, its reducibility is strong, and it is easy to replace H^+^. When Cr appears in an aqueous solution in the form of Cr^2+^, it is in the stable area of Cr^2+^ and H_2_. As a result, hydrogen evolution corrosion can easily occur. Furthermore, HCrO_4_^−^ is located in the high-acid and high-potential area. When Cr exists in the form of CrO_4_^2−^, it appears in a large range above the O_2_ formation, so it is easily oxidized. When Cr exists in an aqueous solution as the Cr(OH)_2_^+^ complex, it mainly exists in a stable state in the passivation region. In an acidic environment with pH < 4.5, Cr also exists in a stable state in the passivation region in the form of Cr^3+^ in a large area. Therefore, when adjusting the Eh and pH values in the leaching solution, attention should be paid to the existing form of Cr in the leaching solution to avoid the negative impact of Cr on the enrichment and separation of the leaching solution.

##### Gibbs Free Energy of Vanadium Bearing Garnet Dissolution Reaction

Vanadium in garnet is mainly in the form of V_2_O_3_ (V (III)), and after being oxidized by an added oxidant, it is mostly in the form of V_2_O_4_ (V (IV)). To explore the mechanism of the enhanced acid-leaching process of garnet, the Gibbs free energy changes of the dissolution reaction of Ca_3_Cr_2_(SiO_4_)_3_ and V_2_O_4_ were calculated. Equations (49)–(54) and (57)–(61) refer to the dissolution reactions of Ca_3_Cr_2_(SiO_4_)_3_ and V_2_O_4_, respectively, whereas Equations (55) and (56) refer to the reaction of adding an oxidant to convert low-valence vanadium into high-valence vanadium.
Ca3Cr2(SiO4)3 + 12H(+a) = 3Ca(+2a) + 2Cr(+3a) + 3H4SiO4(a)(53)
Ca3Cr2(SiO4)3 + 12HF(a) + 12H(+a) = 3Ca(+2a) + 2Cr(+3a) + 3SiF4(g) + 12H2O(l)(54)
Ca3Cr2(SiO4)3 + 14HF(a) + 10H(+a) = 3Ca(+2a) + 2CrF(+2a) + 3SiF4(g) + 12H2O(l)(55)
Ca3Cr2(SiO4)3 + 18HF(a) + 6H(+a) = 3Ca(+2a) + 2CrF3 + 3SiF4(g) + 12H2O(l)(56)
Ca3Cr2(SiO4)3 + 18HF(a) + 6H(+a) = 3Ca(+2a) + 2Cr(+3a) + 3SiF6(−2a) + 12H2O(l)(57)
Ca3Cr2(SiO4)3 + 20HF(a) + 4H(+a) = 3Ca(+2a) + 2CrF(+2a) + 3SiF6(−2a) + 12H2O(l)(58)
V2O3 + MnO2 + 2H+ = V2O4 + Mn2+ + H2O(59)
V2O3 + MnO2 + 4H+ = V2O5 + 2Mn2+ + 2H2O(60)
V2O4 + 4H(+a) + 2SO4(−2a) = 2VOSO4(a) + 2H2O(l)(61)
V2O4 + 2HF(a) + 2H(+a) = 2VOF(+a) + 2H2O(l)(62)
V2O4 + 4HF(a) = 2VOF2(a) + 2H2O(l)(63)
V2O4 + 4HF(a) + 2F(−a) = 2VOF3(−a)+ 2H2O(l)(64)
V2O4 + 4HF(a) + 4F(−a) = 2VOF4(−2a) + H2O(l)(65)

According to the contents of Equations (54)–(64), the relationship between ∆G and T was drawn by using the *HSC Chemistry 9.0* software Reaction Equations module, as shown in Figure 5.

It can be seen from Figure 5a that with increasing temperature from 55 °C to 100 °C, ∆G of the reaction with and without fluorine-bearing leaching aids is less than 0, which indicates that the dissolution reaction of Ca_3_Cr_2_(SiO_4_)_3_ in an acidic environment can be spontaneous. When the temperature is 95 °C, ∆G of the dissolution reaction of Equation (49) (without adding the fluorinating aid) is −241.791 kJ/mol. Meanwhile, ∆G (Equations (54)–(58)) after adding the fluorinating aid decreases from −272.678 kJ/mol to −347.458 kJ/mol, which is 1–1.5 times that when no fluorinating aids are added. This indicates that the dissolution reaction of Ca_3_Cr_2_(SiO_4_)_3_ is more positive when the fluorine-bearing leaching aid is added, and F^-^ promotes Ca_3_Cr_2_(SiO_4_)_3_ to react more thoroughly with the acid, and the V stored in Ca_3_Cr_2_(SiO_4_)_3_ will also be released efficiently.

Figure 5b shows the dissolution reaction after V_2_O_3_ in garnet is oxidized to V_2_O_4_. Obviously, ∆G of all reactions at various temperatures is less than 0, indicating that all reactions can be spontaneous. Among them, the ∆G value with fluorine-bearing leaching aids is higher than that without fluorine-bearing leaching aids, and the difference between the two becomes larger with increasing temperature. Therefore, the addition of fluorine-bearing leaching aids cannot strengthen the leaching of high-valence vanadium in an acidic system, and V_2_O_4_ is more likely to dissolve and produce VOSO_4_ in an environment with only acid.

### 2.2. Leaching

#### 2.2.1. Effect of Sulfuric Acid Dosage on Leaching

The key factor affecting the leaching efficiency of vanadium is the dosage of sulfuric acid. In the acid-leaching process of original samples, the dosage of the leaching agents and acid plays a decisive role in the destruction of aluminosilicate minerals. When the dosage of sulfuric acid is insufficient, the target element V cannot be cracked from the vanadium-bearing mineral lattice, and V cannot be effectively released. When the dosage of sulfuric acid is excessive, the industrial cost increases. This causes difficulties in the purification and enrichment of the subsequent leachate. Therefore, to determine the optimal dosage of sulfuric acid, under the conditions of a leaching time of 5 h, an agitation speed of 500 rpm, leaching aids of 2% + 3% (MnO_2_ + CaF_2_, per 100 g samples; the dosage used in non-enhanced acid leaching is 0%), and a liquid–solid ratio of 1.6:1, the effect of the dosage of sulfuric acid on the leaching efficiency of vanadium was investigated. The results are shown in Figure 6a.

Figure 6a shows that as the dosage of sulfuric acid increases from 17% to 31% (per 100 g of original samples), the leaching efficiency of vanadium with non-enhanced acid leaching increases from 63.39% to 82.03%. Under the same dosage of sulfuric acid, the leaching efficiency of vanadium with enhanced acid leaching greatly increases from 73.92% to 92.11%. When the dosage of sulfuric acid is 28% or more, the leaching efficiency of vanadium exceeds 90%. Compared with non-enhanced acid leaching, enhanced acid leaching not only reduces acid consumption but also achieves excellent leaching efficiency. The higher the dosage of sulfuric acid, the higher the degree of mineral oxidation. Consequently, the reaction of H^+^ with vanadium-bearing minerals is promoted. If the content of sulfuric acid is low, other metal impurities on the mineral surface will consume it first. As a result, sulfuric acid cannot easily penetrate vanadium-bearing minerals, and the reaction with the target minerals is weakened. Therefore, an appropriate dosage of sulfuric acid must be selected to enhance the destruction of the vanadium-bearing mineral structure. However, when the dosage of sulfuric acid is 28%, the leaching efficiency of vanadium reaches 91.86%. Subsequently, the leaching efficiency remains basically unchanged when the dosage of sulfuric acid continues to increase. In summary, 28% sulfuric acid is selected for the intensified acid leaching of original samples.

#### 2.2.2. Effect of Time on Leaching

Generally, if the leaching time is short and the reaction between minerals and leaching aids is incomplete, the vanadium content of the leaching residues will be extremely high, thereby increasing the vanadium loss rate. On the contrary, if the leaching time is extremely high, the production cycle will be extended, production efficiency will be reduced, and production costs will be increased. As a result, the requirements of industrial production will not be met. Therefore, to determine an appropriate leaching time, the effect of time on the leaching efficiency of vanadium was investigated under the conditions of a sulfuric acid dosage of 28%, leaching aids of 2 + 3% (MnO_2_ + CaF_2_), a liquid–solid ratio of 1.6:1, and an agitation speed of 500 rpm. The results are shown in Figure 6b.

As shown in Figure 6b, when the leaching time is extended from 2 h to 6 h, the leaching efficiency of vanadium with non-enhanced acid leaching increases slowly from 71.03% to 81.89% (by only 10.86%, which is a small increase). The reason for the low leaching efficiency is that vanadium is mostly stored in aluminosilicate minerals in the form of a trivalent. In addition, the crystal structure is stable, and the cracking effect of the target minerals is poor only by sulfuric acid leaching. Therefore, to obtain a higher vanadium leaching index in this system, the leaching time should be increased.

When the leaching time is 2 h, the leaching efficiency of vanadium using enhanced acid leaching is 9.92% higher than that using non-enhanced acid leaching. When the time is extended to 6 h, the leaching efficiency using enhanced acid leaching is 10.47% higher than that using non-enhanced acid leaching. It can also be seen from Figure 6b that when the leaching time reaches 4 h, the leaching efficiency of vanadium can be as high as 91.86% or more. If the leaching time is extended for 1–2 h, the leaching efficiency of vanadium maintains a consistent state. Therefore, on the premise of ensuring the vanadium leaching index and considering the industrial production cost, the leaching time of 4 h is selected. Based on the leaching results of these two systems, the leaching efficiency of vanadium in the enhanced acid-leaching system with leaching aids is approximately 10% higher than that in the non-enhanced acid-leaching system, ensuring production efficiency.

#### 2.2.3. Effect of Liquid–Solid Ratio on Leaching

If the liquid–solid ratio is extremely high, the vanadium concentration in the leaching solution will be reduced, which complicates subsequent purification and enrichment. If the leaching solution is extremely low, the pulp concentration is too high, which will affect the mass transfer and diffusion of the pulp. Therefore, the liquid–solid ratio is one of the key factors affecting the leaching process. To ensure the purification and enrichment of the subsequent leaching solution, it is crucial to select a suitable liquid–solid ratio. Therefore, under the conditions of a sulfuric acid dosage of 28%, leaching aids of 2 + 3% (MnO_2_ + CaF_2_), a leaching time of 4 h, and an agitation speed of 500 rpm, the effect of the liquid–solid ratio on the leaching efficiency of vanadium was investigated. The results are shown in Figure 6c.

It can be seen from Figure 6c that the liquid–solid ratios of both leaching systems first increase and then decrease slowly with leaching. When the liquid–solid ratio is increased from 0.7:1 to 1.3:1, the leaching efficiency increases from 80.12% to 82.53% with non-enhanced acid leaching. Meanwhile, the leaching efficiency decreases gradually when the liquid–solid ratio is increased to 1.6:1. The reasons for this phenomenon may be that the liquid–solid ratio is low, the slurry becomes sticky and thick, some minerals are not immersed completely in the solution during the leaching process, and the stirring resistance is also increased. Therefore, the liquid–solid ratio must be increased to make the minerals disperse and contact fully and promote the reaction in the solution. However, when the liquid–solid ratio continues to increase, the H^+^ concentration in the pulp decreases, thereby hindering the destruction of the crystal structure of aluminosilicate minerals. Consequently, vanadium leaching becomes difficult. When the liquid–solid ratio is 1:1, the maximum leaching efficiency reaches 93.52%. However, the leaching efficiency starts to decline as the liquid–solid ratio continues to increase. Therefore, the optimal liquid–solid ratio is 1:1 for enhanced acid leaching.

#### 2.2.4. Effect of MnO_2_ Dosage on Leaching

The addition of MnO_2_ can oxidize the low-valent vanadium of aluminosilicate minerals and improve the leaching efficiency of vanadium. To investigate the effect of the leaching aids, under the conditions of a sulfuric acid dosage of 28%, the content of fluorine-bearing leaching aids of 3% (CaF_2_), an agitation speed of 500 rpm, a leaching time of 4 h, and a liquid–solid ratio of 1:1, the MnO_2_ dosage condition test was conducted. The results are shown in Figure 6d.

It can be seen from Figure 6d that, when MnO_2_ is not added, the leaching efficiency of a single leaching aid (CaF_2_) is poor, leaching efficiency is only 85.17%, and leaching efficiency continues to increase with the dosage of MnO_2_. This is because MnO_2_ has strong oxidation, weakens the Al–O bond, destroys the surface of vanadium-bearing minerals, and facilitates the leaching of vanadium. When the dosage of MnO_2_ exceeds 1%, the leaching efficiency of vanadium will decline slowly. This may be due to the simultaneous catalytic effect of MnO_2_. Too high a dosage of MnO_2_ will reduce the leaching efficiency, cause waste of reagents, and increase unnecessary production costs. Therefore, the optimal dosage of MnO_2_ is 1%.

#### 2.2.5. Effect of CaF_2_ Dosage on Leaching

Fluorine-bearing leaching aids can make Al and Si in aluminosilicate form a complex with F, force the Al–O bond breaking, and strengthen vanadium leaching. Under the conditions of a sulfuric acid dosage of 28%, an oxidant of 1% (MnO_2_), an agitation speed of 500 rpm, a leaching time of 4 h, and a liquid–solid ratio of 1:1, the condition test of the CaF_2_ dosage was carried out. The results are shown in Figure 6e.

It can be seen from Figure 6e that when CaF_2_ is not added, the leaching efficiency reaches 85.15%. With the addition of fluoride, the reaction with Si strengthens the acid-leaching process and accelerates the cracking of vanadium-bearing mineral particles. Therefore, when the dosage of CaF_2_ increases from 1% to 4%, the leaching efficiency of vanadium also continues to increase. With increasing the dosage of CaF_2_, the destruction of vanadium-bearing minerals is enhanced, and vanadium is more easily dissolved from the minerals. Thus, the leaching efficiency of vanadium is improved. When the dosage of CaF_2_ reaches 3%, the leaching efficiency basically maintains a balanced state. Considering the industrial production cost, the dosage of CaF_2_ is selected as 3%.

Some studies [6] have shown that the leaching efficiency is higher when two types of leaching aids are added simultaneously than when they are added individually. Therefore, two types of leaching aids were added simultaneously in this experiment.

#### 2.2.6. Effect of Temperature on Leaching

The effect of temperature on leaching efficiency is obvious. Increasing the temperature is beneficial to the leaching reaction. However, it is meaningless to increase the leaching efficiency when the temperature is increased to a certain extent because it will increase energy consumption and lead to higher production costs. Therefore, under the conditions of a sulfuric acid dosage of 28%, an agitation speed of 500 rpm, leaching aids of 1 + 3% (MnO_2_ + CaF_2_), a liquid–solid ratio of 1:1, and a leaching time of 4 h, the influence of temperature on leaching efficiency of vanadium was investigated. The results are shown in Figure 6f.

According to Figure 6f, the leaching efficiency is directly proportional to the increase in the temperature, and the leaching efficiency under enhanced acid leaching is approximately 10% higher than that under non-enhanced acid leaching. Moreover, the leaching efficiency of vanadium is approximately 81% when no leaching aid is added, whereas it is approximately 85% when only a single leaching aid is added. Therefore, the leaching efficiency of vanadium can be increased by approximately 10% when the 2 leaching aids are added for enhanced acid leaching compared with that under non-enhanced acid leaching. When the temperature increases from 55 °C to 95 °C, the leaching efficiency of vanadium with non-enhanced acid leaching increases from 58.27% to 83.41%. However, the leaching efficiency increases from 69.47% to 94.86% in the case of enhanced acid leaching. Because the increase in the temperature is conducive to increasing the energy inside the mineral particles and the ability to crack the mineral particles enhances, the activity of the leaching system will be stimulated, with the increase in the temperature, to promote the positive direction of the leaching reaction. Furthermore, the increase in the temperature helps accelerate the thermal movement of the solution molecules. Consequently, the diffusion rate of the mineral particles is improved, and H^+^ can easily enter the particles to weaken the chemical bonds of the minerals and destroy their structures. Hence, the cracking speed of the mineral particles is accelerated, and the vanadium leaching index is improved. Because the boiling water bath at the test site was at 95 °C, the continuous temperature increase had little effect on the leaching efficiency. Finally, the selected leaching temperature is 95 °C.

The continuous experiment was carried out according to the optimum conditions, and the experimental parameters were adjusted according to the results. Finally, the leaching efficiency of vanadium in the enhanced acid-leaching system is about 94.86%, which is about 11.45% higher than that of the unreinforced acid-leaching system. Among them, the dosage of sulfuric acid is 28%, leaching time is 4 h, liquid–solid ratio is 1:1, the dosage of the leaching aids is 1 + 3%, and the leaching temperature is 95 °C. Furthermore, under the same leaching conditions, it is strongly verified that enhanced acid leaching can improve the leaching efficiency and performance of the leaching system compared with non-enhanced acid leaching.

### 2.3. XRD Analysis

Figure 7 analyzes the changes in the mineral phases before and after leaching. Among them, (a), (b), and (c) represent the XRD patterns of the original samples, non-enhanced acid-leaching residues, and enhanced acid-leaching residues, respectively. It can be seen from Figure 7 that the main phases of the original samples are quartz (PDF#85-0930), mica (PDF#88-0971), illite (PDF#86-2237), and limonite (PDF#79-0419). Among them, limonite may be the product of the dehydration of iron-bearing minerals in the samples during vacuum drying after oxidation. After leaching, the characteristic diffraction peaks of mica, illite, and limonite show significantly weakened states when acid leaching is non-enhanced, and the intensity of the mica peak is reduced by half. When acid leaching is enhanced, the characteristic diffraction peaks of mica and illite are significantly weakened compared with those of non-enhanced acid leaching, the characteristic diffraction peaks of limonite disappear, and the peak intensity of illite is reduced by about 1/3 compared with the unreinforced acid leaching. This shows that the acid can damage the mineral structure to a certain extent under the condition of non-enhanced acid leaching, but the damage is lower. In the case of enhanced acid leaching, leaching aids can help acids destroy the mineral structure and promote leaching, thus improving the leaching efficiency of vanadium.

Crucially, illite in the original sample does not contain vanadium, which is inconsistent with the existing literature report that “vanadium in most vanadium-bearing shale exists in illite”. Therefore, taking the original sample as the research object, which is different from the conventional siliceous vanadium-bearing shale, has distinct significance.

### 2.4. AFM Analysis

Two-dimensional and three-dimensional images and cross-section height images of sample surfaces in nano-scale were obtained using the AFM imaging technology, and they were used to analyze whether the leaching aids enhanced the mineral leaching process. Figure 8a–i show the changes in the cross-section height of the original samples, non-enhanced acid-leaching residues, and enhanced acid-leaching residues, respectively. Consequently, different acid–mineral reaction degrees caused by the presence or absence of the leaching aids can be determined by changes in the mineral surface morphology and cross-section height.

According to Figure 8c, the cross-section height map of the original samples shows that the peak value is at 20 nm. It can also be seen from Figure 8a that the surface of the original samples is relatively smooth and flat. In Figure 8d, a cylindrical pattern with a large cross-sectional area is present. It is probably caused by a large particle in the AFM-imaging scanning area. Therefore, the height of the cross-section shown in Figure 8f has a convex peak. Owing to the difference in the scanning range, the corresponding error will be brought. Thus, the convex peak can be ignored. The height of the cross-section is considered as 3 nm when the acid leaching is not strengthened. The surface of leaching residues after enhancing acid leaching shown in Figure 8g is the roughest. Similarly, it can be seen from Figure 8i that its cross-section height (1.5 nm) is considerably smaller than the previous 2 values. This shows that the addition of leaching aids promotes the interaction between the acid and the mineral surface, and the cross-section height of the mineral surface decreases gradually.

The results of the AFM confirmed that the cross-section height of each leaching system was reduced correspondingly compared with that of the original samples after both the enhanced and non-enhanced acid-leaching processes. Thus, the leaching process of both systems was effective. However, when acid leaching was enhanced, the reaction degree between the minerals and the acid was the highest. This is because the addition of the leaching aids strengthened the leaching process and promoted the leaching effect.

### 2.5. FT-IR Analysis

The results of AFM analysis confirmed that the acid reacted with the mineral surface, and the reaction intensity was higher after adding the leaching aids. This indicated that the leaching aids could change the mineral surface properties. Therefore, to study the adsorption of the leaching aids on the mineral surface, the specific spectral peaks of the mineral and leaching aids were analyzed by FT-IR to confirm that the leaching aids enhanced the acid-leaching process. Figure 9a, Figure 9b and Figure 9c, respectively, represent the FT-IR spectra of the original samples, non-enhanced acid-leaching residues, and enhanced acid-leaching residues. In Figure 9a, a strong absorption peak appears at a wave number of 1087.80 cm^−1^, which is caused by the antisymmetric stretching vibration of the Si (Al)–O–Si (Al) bond. The absorption peak at 513 cm^−1^ is attributed to the bending vibration coupling of Si–O–Al and Si–O. After non-enhanced acid leaching and enhanced acid leaching, the Si (Al)–O–Si (Al) stretching vibration absorption band at 1087.80 cm^−1^ in the original samples moves to 1089.73 cm^−1^ and 1095.52 cm^−1^ in the high-frequency direction, respectively. Furthermore, the absorption strength increases with the addition of the leaching aids. This shows that the Si–O tetrahedron in the muscovite is constantly deformed, and the Al–O octahedron structure loses its original stability. Crucially, the absorption intensity of the 513 cm^−1^ absorption peak in spectrum (a) is weakened in spectrum (b) and completely disappears in spectrum (c). The characteristic absorption peak of CaSO_4_ also appears at 599.83 cm^−1^ in spectrum (c). All these results confirmed that the fluorine-containing leaching aids can react in the leaching solution, which is consistent with the results of the thermodynamic calculation. This also shows that the stability of the Si–O tetrahedron and Al–O octahedron structure in muscovite seriously deteriorates during acid leaching containing F^-^ until they are completely destroyed. FT-IR spectrum analysis confirmed that enhanced acid leaching can reduce Si–O and Al–O chemical bonds in vanadium-bearing mica. It is inferred that enhanced acid leaching can effectively dissolve vanadium and improve leaching performance, which is consistent with the leaching experiment results.

### 2.6. SEM Analysis

The results of the FT-IR analysis show that the fluorine-bearing leaching aids can be adsorbed on the mineral surface and thus helping H^+^ to enhance the leaching process. Therefore, for a more intuitive understanding of the changes in the mineral surface morphology caused by leaching aids, SEM analysis was carried out on original samples, non-enhanced acid-leaching residues, and enhanced acid-leaching residues. The results are shown in Figure 10a, Figure 10b and Figure 10c, respectively.

In Figure 10a, it can be seen that the surface structure of the original samples is extremely tight. As shown in Figure 10b, the surface structure of non-enhanced acid-leaching residues is loose and has a few gaps. This may be because the acid is not completely dispersed on the surface of the vanadium-bearing minerals to react with it during the leaching process. Consequently, only a small area of the mineral surface is corroded by the acid. Therefore, when acid leaching is not strengthened, the leaching effect is poor. The SEM image of leaching residues after adding the leaching aids to enhance acid leaching is shown in Figure 10c; there are many pores on the surface of the leaching residues, and the whole structure is very loose. Therefore, H^+^ is easily diffused to the interior of the mineral to strengthen the leaching process, and the leaching efficiency of vanadium increases compared with non-enhanced acid leaching.

The results of SEM image analysis and thermodynamic calculation can infer that enhanced acid leaching can help H^+^ strengthen the erosion of the mineral surface, making the mineral surface rougher and showing a porous and loose state. Therefore, H^+^ can quickly flow into the mineral through the gap, make contact with the mineral in a large area, and completely react with the vanadium-bearing mineral. Therefore, vanadium can be effectively dissolved, which can not only improve the leaching efficiency of vanadium but also improve the leaching performance.

## 3. Experimental

### 3.1. Materials and Reagents

The used original samples (acid leaching feed) were from the flotation preconcentration process of vanadium-bearing shale (the content of V_2_O_5_ in vanadium-bearing shale is 0.83%) in Shangluo, Shaanxi, China. For the three-stage open circuit procedure of the pre-concentration process, the used reagents were sulfuric acid (pH regulator), sodium silicate (dispersant), sodium fluosilicate (inhibitor), and dodecylamine (collector). The main chemical components, valence distribution, and phase of the original samples are shown in Table 4, Table 5 and Table 6, respectively.

Table 4 shows that the content of V_2_O_5_ in flotation concentrate is 1.96%, which indicates that flotation greatly improves the grade of V_2_O_5_ and has an obvious preconcentration effect. Nowadays, the high cost of vanadium extraction is mostly due to the low grade of vanadium. The flotation preconcentration process is undoubtedly of great significance to improve the vanadium grade and reduce the cost of vanadium extraction. Additionally, the original samples are based on SiO_2_, with the content of SiO_2_ as high as 86.49%, followed by an Al_2_O_3_, which grade is 3.78%. According to this, the original samples should contain a large dosage of quartz (SiO_2_) and aluminosilicate minerals. Additionally, alkalinity is R = (CaO + MgO)/(SiO_2_ + Al_2_O_3_) = 0.021. Therefore, from the alkalinity calculation results, it is concluded that the original samples belong to acid ore and are suitable for extracting vanadium through acid leaching.

Table 5 shows that most of the vanadium is in the form of low valence V (III), with the distribution rate as high as 74.55%, whereas the distribution rates of V (IV) and V (V) are very low, 22.73% and 2.72%, respectively. However, V (III) mainly occurs in the lattice of aluminosilicate minerals, replacing Al (III) in six-times-coordinated Al–O octahedron in the form of isomorphism and entering the mineral lattice [24]. V (III) is restrained by the stable layered aluminosilicate crystal structure. Its state is particularly stable, making it extremely difficult to dissolve in acid and aqueous solutions. Only by breaking the Al–O bond in the lattice of aluminosilicate minerals can vanadium be released and then oxidized into high-valence vanadium, which can form vanadium that is easily dissolved in acid and aqueous solutions. Therefore, in the acid-leaching process, it is necessary to break the mineral lattice and convert low valence of vanadium into high valence of vanadium.

Table 6 shows that vanadium is mainly hosted by muscovite (49.36%), garnet (21.10%), and biotite (17.26%). A lower dosage of vanadium occurs in iron oxide (3.81%) and chlorite (3.03%). In addition, trace vanadium is dispersed in wollastonite, ilmenite, and V_2_O_5_.

Among them, garnet is an island aluminosilicate mineral, which is difficult to dissolve in acidic or aqueous solution. Therefore, it is difficult to extract vanadium under the conditions of non-roasting and atmospheric pressure. In order to extract vanadium from insoluble aluminosilicate and improve the leaching efficiency of vanadium and, subsequently, the process of vanadium extraction, research on enhanced acid leaching must be discussed.

In these experiments, the following reagents were used: sulfuric acid (H_2_SO_4_, 98%), manganese dioxide (MnO_2_, 1%), and calcium fluoride (CaF_2_, 3%), which were, respectively used as a leaching agent or pH regulator, an oxidant, and a fluorinating aid (leaching aid). The sulfuric acid was purchased from Chengdu Kelong Chemical Reagent Factory, and the leaching aids were purchased from Tianjin Fengchuan Chemical Reagent Technology Co., Ltd. All the reagents were of analytical grade.

### 3.2. Thermodynamics

Thermodynamics mainly includes checking the main mineral phases in leaching reaction balance, the state and conditions that exist with the changes of energy to determine the direction and limit of chemical reaction. Thermodynamic calculations are of great value in analyzing the equilibrium conditions of each component (solid, liquid, and gas) in the leaching process of hydrometallurgy. *HSC 9.0 software* was used to calculate the thermodynamic parameters of the reactions of the main vanadium-bearing mineral phases (vanadium-bearing mica and garnet) in the acid-leaching system of the original samples. Both the mathematical derivation and thermodynamic principles were used, and the Eh-pH diagram of the enhanced acid-leaching system was drawn to describe the thermodynamics of the interaction of metal-H_2_O to clarify the balance of mineral-H_2_O. The relationship between the change in the standard Gibbs free energy (∆G) and the temperature (T) of the dissolution reaction of the main minerals during enhanced and non-enhanced acid-leaching processes was analyzed. As a result, a theoretical basis for the spontaneity of dissolution reaction of vanadium-bearing minerals in flotation concentrate was provided.

### 3.3. Leaching

The thermostat water bath cauldron was preheated before leaching experiments, and 100 g of original samples were uniformly sampled each time and placed into a beaker (500 mL). A certain dosage of leaching aids, H_2_O, and H_2_SO_4_, were added into the beaker, thoroughly mixed, and then placed into the thermostat water bath cauldron. After leaching experiments, the pulp was filtered, dried, and weighed. The volume and concentration of the leaching solution were obtained to calculate the leaching efficiency. Each group of experiments was repeated thrice, and the results were averaged.

### 3.4. XRD Analysis

XRD analysis is an important widely used method to analyze mineral phases. Its measuring *2θ* arrange from 10° to 80°, the step length is 0.022°, and the time of each step is 0.2 s. The samples of −37 μm were used to prepare the XRD pattern. Based on the information from the diffraction spectrogram, the mineral phases under the conditions of the original samples, enhanced acid leaching, and non-enhanced acid leaching were observed, and the presence of dislocations or lattice defects in the mineral interior was also investigated to analyze the changes in the microstructure of samples before and after leaching.

### 3.5. AFM Analysis

AFM is usually used to describe the surface properties of minerals through imaging technology. The height of the surface cross-section of the leaching residues under the conditions of the original samples, enhanced acid leaching, and non-enhanced acid leaching were observed by AFM (*Dimension Icon, Bruker AXS, Germany, Beijing Representative Office, China*). The scanning range was 5 μm × 5 μm, and images obtained by AFM were analyzed using the *NanoScope 1.5 software* to reveal the changes in the mineral surface morphology and cross-section height under different action conditions. Before scanning, the three samples were prepared by water dispersion, and each action condition was consistent with the best parameters of leaching experiments.

### 3.6. FT-IR Analysis

FT-IR was used to characterize the surface reagent adsorption before and after the interaction of minerals and reagents. Mineral samples with a particle size of −5μm were used to prepare infrared spectrum. The preparation procedure was the same as leaching experiments. After the preparation, the leaching solution was filtered with a vacuum filter to obtain a filter cake. Subsequently, a vacuum drying oven was used to dry the filter cake at room temperature (20–25 °C). In a natural agate mortar, 10 mg mineral samples and 500 mg KBr were mixed evenly each time, and then the powder was pressed into a thin round cake under 12 Mpa. The round cake samples were placed on a *Nicolet iS 10 spectrometer* (Thermo Fisher, Waltham, MA, USA) to measure the FT-IR. The range was 400–4000 cm^−1^. Scanning was performed 50 times, and the spectrum was recorded with a 16 cm^−1^ resolution. The experiment was repeated thrice.

### 3.7. SEM Analysis

SEM is often used to observe changes in the mineral surface micromorphology. First, three samples of the original samples, non-enhanced acid leaching residues, and enhanced acid leaching residues, all of them with a particle size of −37 μm, were sprayed with spray-gold treatment for 45 s. Subsequently, the samples were photographed with a *Czech TESCAN MIRA LMS electron microscope*, Philips, Netherlands. The accelerating voltage of the instrument was 3 kV, and the selected magnification was ×10,000. Finally, the high-energy electron beam of SEM was targeted at the samples to stimulate a variety of physical information. Through reception, amplification, and display imaging of the physical information, the mineral surface morphology could be observed.

## 4. Conclusions

In this study, the mechanism of the mineral microstructure evolution of the flotation concentrate of vanadium-bearing shale during acid leaching caused by leaching aids was revealed. MnO_2_ can oxidize low-valence vanadium to high-valence vanadium, whereas CaF_2_ can break the Al–O and Si–O bonds in vanadium-bearing minerals. Both can enhance vanadium leaching. Thermodynamic calculations show that reactions can be spontaneous regardless of whether the leaching aids are added. Furthermore, the leaching aids can completely destroy the lattice structure of insoluble minerals such as garnet, thus strengthening the vanadium leaching. The results of leaching experiments show that the leaching aids can strengthen the acid-leaching process of flotation concentrate and improve the leaching efficiency of vanadium. XRD analysis shows that the leaching aids in addition to acid leaching can weaken the intensity of the characteristic diffraction peaks of quartz, illite, and other minerals in samples. AFM analysis confirmed that the leaching aids can stimulate the acid–mineral reaction degree. FT-IR analysis shows that Al–O and Si–O in the mica lattice are broken, and the mineral structure is destroyed when the leaching aids are present. Finally, it was observed by SEM that the mineral surface structure becomes loose and porous after adding the leaching aids, and enhanced acid leaching promotes the acid–mineral reaction in a large area. Overall, this work is helpful to understand the mechanism of enhanced acid leaching of the flotation concentrate of vanadium-bearing shale. Based on the mechanism study of enhanced acid leaching, we hope to continue to develop a novel process with green, efficient, and high practical value to extract vanadium from the flotation concentrate of vanadium bearing shale in the future.

## Figures and Tables

**Figure 1 molecules-28-02706-f001:**
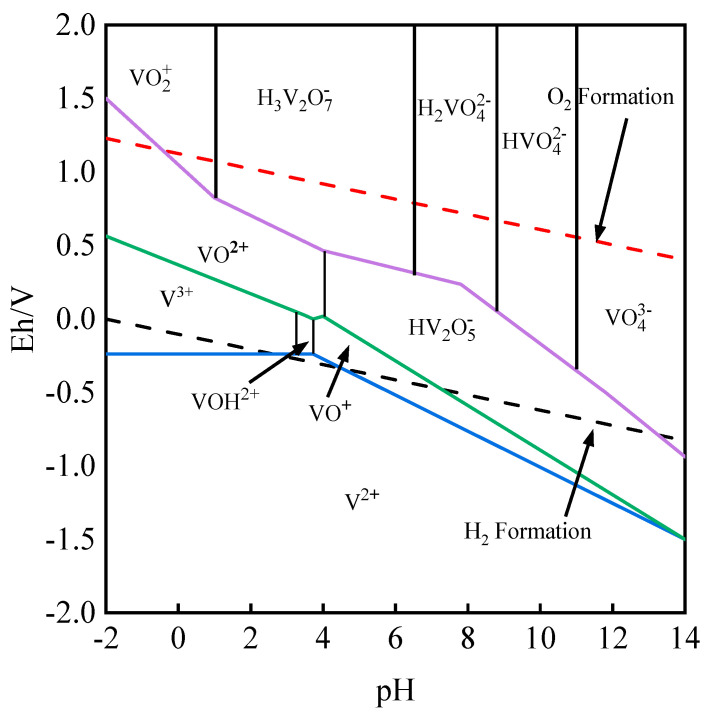
Eh–pH relationship of V-H_2_O system.

**Figure 2 molecules-28-02706-f002:**
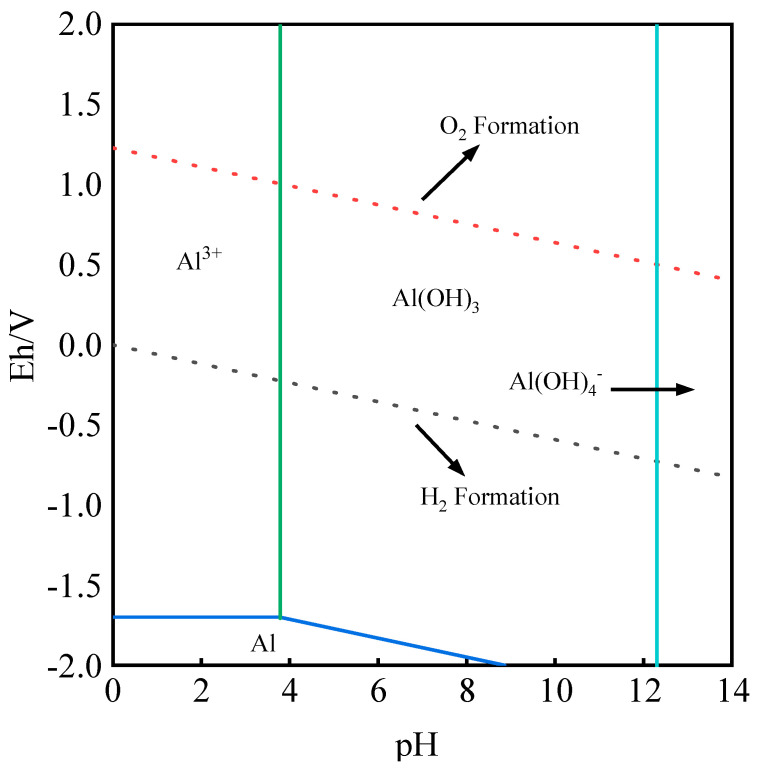
Eh–pH relationship of Al-H_2_O system.

**Figure 3 molecules-28-02706-f003:**
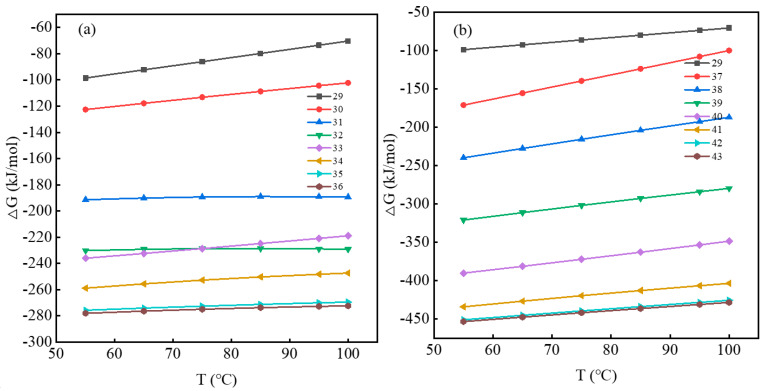
∆G change curve with T: (**a**) Equations (29)–(36); (**b**) Equations (29) and (37)–(43).

**Figure 4 molecules-28-02706-f004:**
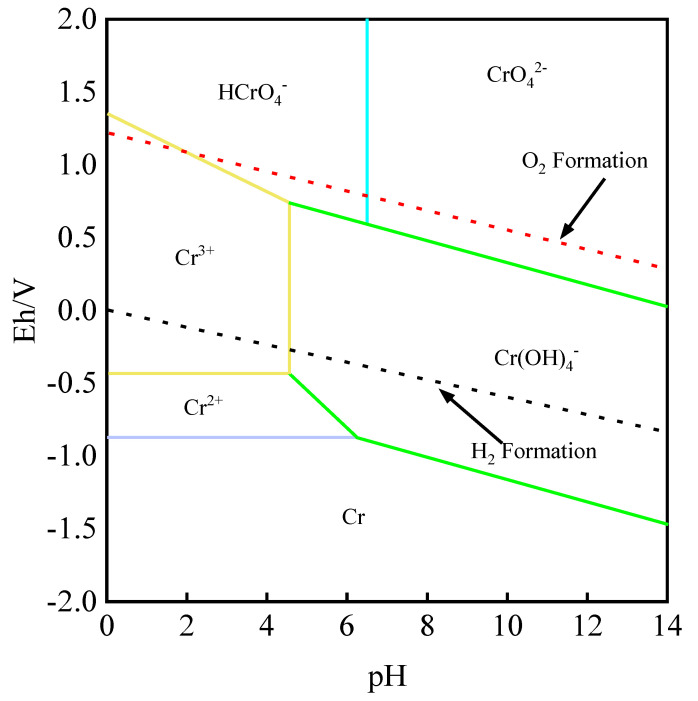
Eh–pH relationship of Cr-H_2_O system.

**Figure 5 molecules-28-02706-f005:**
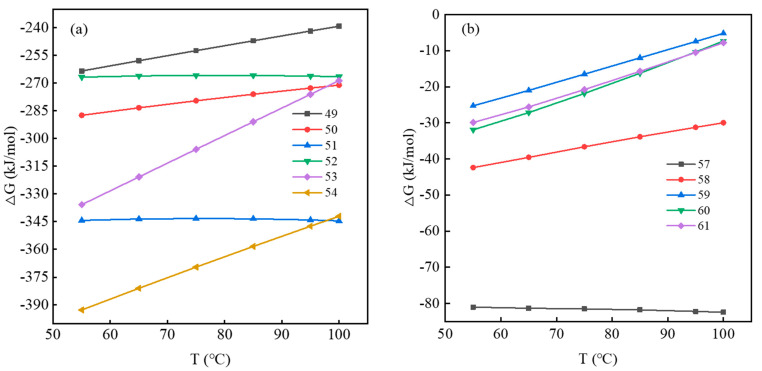
∆G change curve with T: (**a**) Equations (53)–(58); (**b**) Equations (61)–(65).

**Figure 6 molecules-28-02706-f006:**
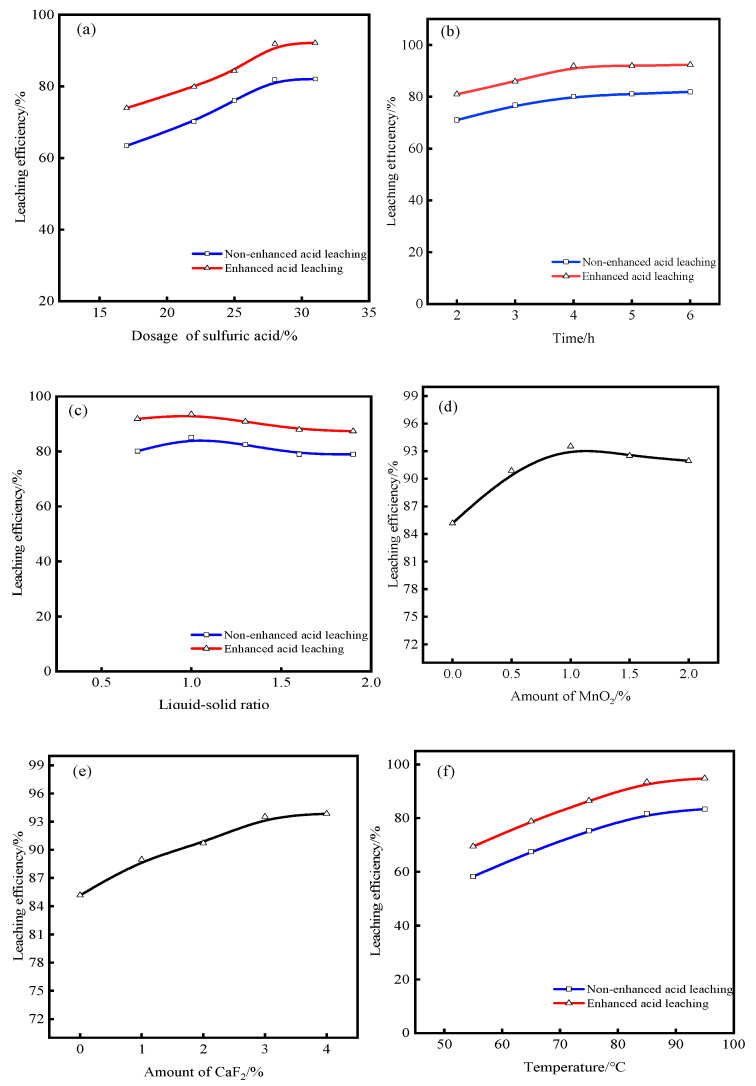
Factors affecting leaching efficiency: (**a**) dosage of sulfuric acid; (**b**) time; (**c**) liquid–solid ratio; (**d**) dosage of MnO_2_; (**e**) dosage of CaF_2_; and (**f**) temperature.

**Figure 7 molecules-28-02706-f007:**
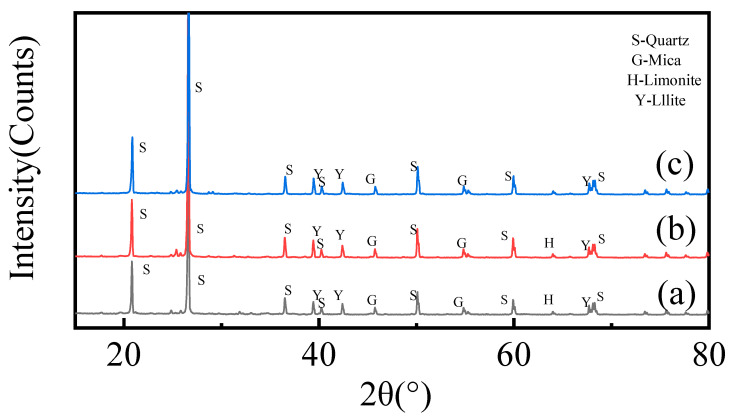
XRD spectrum: (**a**) original samples; (**b**) non-enhanced acid-leaching residues; and (**c**) enhanced acid-leaching residues.

**Figure 8 molecules-28-02706-f008:**
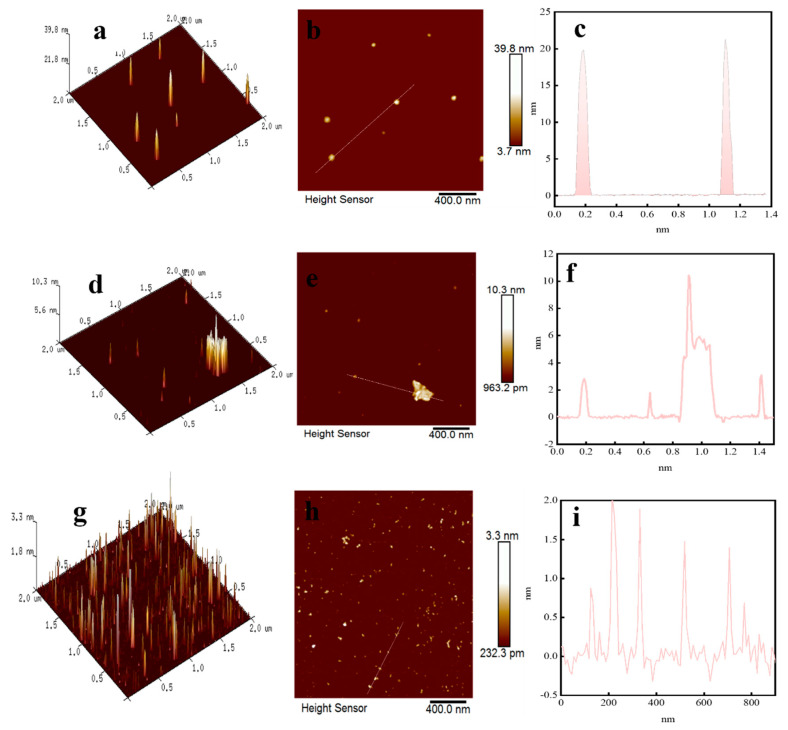
AFM image: (**a**–**c**) original samples; (**d**–**f**) non-enhanced acid-leaching residues; and (**g**–**i**) enhanced acid-leaching residues.

**Figure 9 molecules-28-02706-f009:**
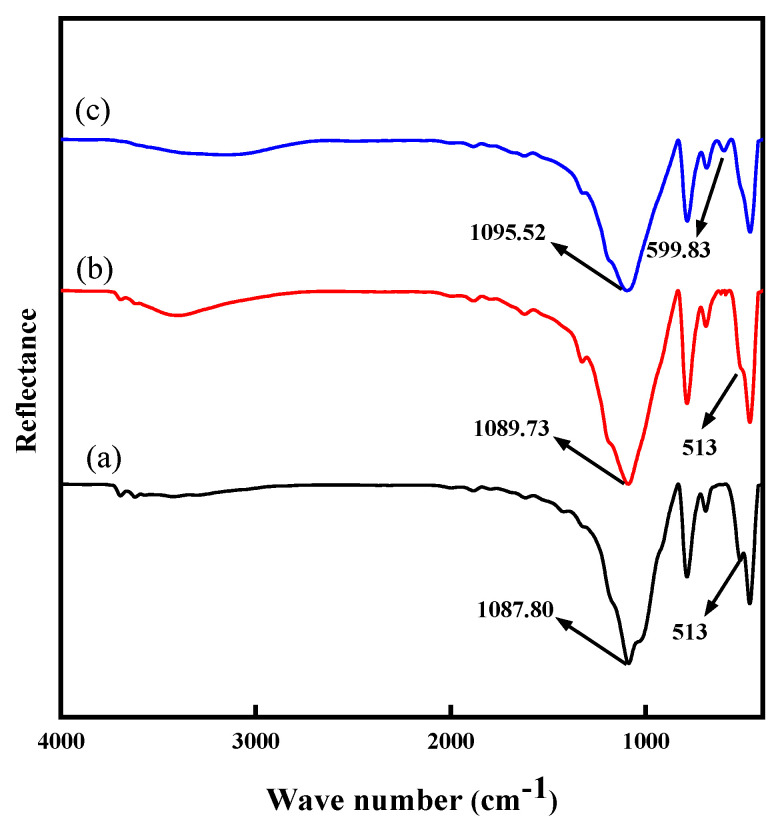
FT-IR spectrum: (**a**) original samples; (**b**) non-enhanced acid-leaching residues; and (**c**) enhanced acid-leaching residues.

**Figure 10 molecules-28-02706-f010:**
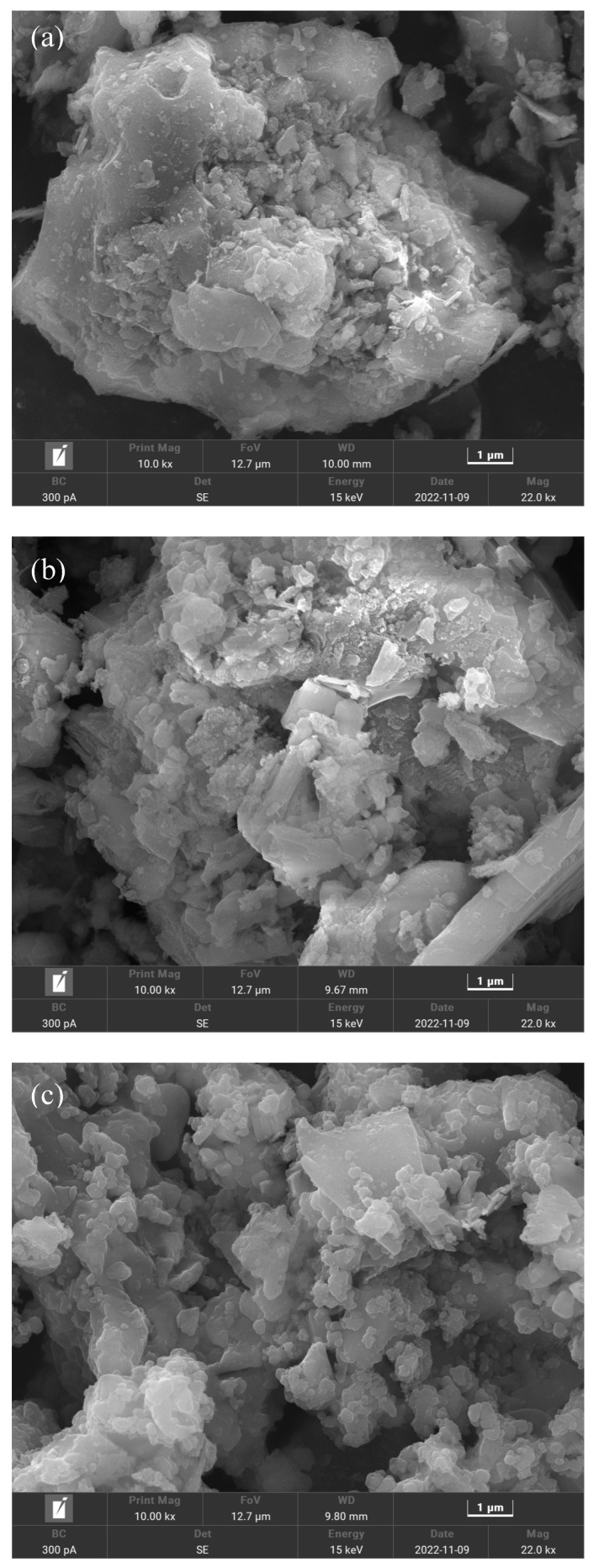
SEM image: (**a**) original samples; (**b**) non-enhanced acid-leaching residues; and (**c**) enhanced acid-leaching residues.

**Table 1 molecules-28-02706-t001:** Main electrochemical reaction equations of V-H_2_O system.

Electrochemical Reaction Equation	Eh-pH	Number
VOH^2+^ = VO^+^ + H^+^	$\mathrm{pH}=3.52+\mathrm{lg}\frac{\left({\mathrm{VO}}^{+}\right)}{\left({\mathrm{VOH}}^{2+}\right)}$	(5)
V^3+^ + H_2_O = VOH^2+^ + H^+^	$\mathrm{pH}=2.92+\mathrm{lg}\frac{\left({\mathrm{VOH}}^{2+}\right)}{\left(V^{3+}\right)}$	(6)
2VO^2+^ + 3H_2_O = HV_2_O_5_^−^ + 5H^+^	$\mathrm{pH}=4.024+1/5\mathrm{lg}\frac{\left({\mathrm{HV}}_{2}O_{5}^{-}\right)}{\;{\left({\mathrm{VOH}}^{2+}\right)}^{2}}$	(7)
2VO^2+^ + 3H_2_O = H_3_V_2_O_7_^−^ + 3H^+^	$\mathrm{pH}=1.03+1/3\mathrm{lg}\frac{\left(H_{3}V_{2}O_{7}^{-}\right)}{\;{\left({\mathrm{VO}}_{2}{}^{+}\right)}^{2}}$	(8)
H_3_V_2_O_7_^−^ + H_2_O = 2H_2_VO_4_^−^ + 2H^+^	$\mathrm{pH}=7.38+\mathrm{lg}\frac{\left(H_{2}{\mathrm{VO}}_{4}^{-}\right)}{\left(H_{3}V_{2}O_{7}^{-}\right)}$	(9)
H_2_VO_4_^−^ = HVO_4_^2−^ + H^+^	$\mathrm{pH}=9.52+\mathrm{lg}\frac{\left({\mathrm{HVO}}_{4}^{2-}\right)}{\left(H_{2}{\mathrm{VO}}_{4}^{-}\right)}$	(10)
HVO_4_^2−^ = VO_3_^3−^ + H^+^	$\mathrm{pH}=11.50+\mathrm{lg}\frac{\left({\mathrm{VO}}_{3}^{3-}\right)}{\left({\mathrm{HVO}}_{4}^{2-}\right)}$	(11)
V^3+^ + *e*^−^ = V^2+^	$E=-0.255+0.0591\mathrm{lg}\frac{\left(V^{3+}\right)}{\left(V^{2+}\right)}$	(12)
VO^+^ = VO^2+^ + *e*^−^	$E=-0.044+0.0591\mathrm{lg}\frac{\left({\mathrm{VO}}^{2+}\right)}{\left({\mathrm{VO}}^{+}\right)}$	(13)
V^2+^ + H_2_O = VOH^2+^ + H^+^ + *e*^−^	$E=-0.082-0.0591\mathrm{pH}+0.0591\mathrm{lg}\frac{\left({\mathrm{VOH}}^{2+}\right)}{\left(V^{2+}\right)}$	(14)
V^2+^ + H_2_O = VO^+^ + 2H^+^ + *e*^−^	$E=0.126-0.1182\mathrm{pH}+0.0591\mathrm{lg}\frac{\left({\mathrm{VO}}^{+}\right)}{\left(V^{2+}\right)}$	(15)
V^3+^ + H_2_O = VO^2+^ + 2H^+^ + *e*^−^	$E=0.337-0.1182\mathrm{pH}+0.0591\mathrm{lg}\frac{\left({\mathrm{VO}}^{2+}\right)}{\left(V^{3+}\right)}$	(16)
VOH^2+^ = VO^2+^ + H^+^ + *e*^−^	$E=0.164-0.0591\mathrm{pH}+0.0591\mathrm{lg}\frac{\left({\mathrm{VO}}^{2+}\right)}{\left({\mathrm{VOH}}^{2+}\right)}$	(17)
2VO^+^ + 3H_2_O = HV_2_O_5_^−^ + 5H^+^ + 2*e*^−^	$E=0.551-0.1477\mathrm{pH}+0.0295\mathrm{lg}\frac{\left({\mathrm{HV}}_{2}O_{5}^{-}\right)}{{\left({\mathrm{VO}}^{+}\right)}^{2}}$	(18)
VO^2+^ + H_2_O = VO_2_^+^ + 2H^+^ + *e*^−^	$E=1.004-0.1182\mathrm{pH}+0.0591\mathrm{lg}\frac{\left({\mathrm{VO}}_{2}^{+}\right)}{\left({\mathrm{VO}}^{2+}\right)}$	(19)
2VO^2+^ + 5H_2_O = H_3_V_2_O_7_^−^ + 7H^+^ + 2*e*^−^	$E=1.096-0.2068\mathrm{pH}+0.0295\mathrm{lg}\frac{\left(H_{3}V_{2}O_{7}^{-}\right)}{{\left({\mathrm{VO}}^{2+}\right)}^{2}}$	(20)
HV_2_O_5_^−^ + 2H_2_O = H_3_V_2_O_7_^−^ + 2H^+^ + 2*e*^−^	$E=0.501-0.0591\mathrm{pH}+0.0295\mathrm{lg}\frac{\left(H_{3}V_{2}O_{7}^{-}\right)}{\left({\mathrm{HV}}_{2}O_{5}^{-}\right)}$	(21)
HV_2_O_5_^−^ + 3H_2_O = 2H_2_VO_4_^−^ + 3H^+^ + 2*e*^−^	$E=0.719-0.0886\mathrm{pH}+0.0295\mathrm{lg}\frac{{\left(H_{2}{\mathrm{VO}}_{4}^{-}\right)}^{2}}{\left({\mathrm{HV}}_{2}O_{5}^{-}\right)}$	(22)
HV_2_O_5_^−^ + 3H_2_O = 2HVO_4_^2−^ + 5H^+^ + 2*e*^−^	$E=1.281-0.1477\mathrm{pH}+0.0295\mathrm{lg}\frac{{\left({\mathrm{HVO}}_{4}^{2-}\right)}^{2}}{\left({\mathrm{HV}}_{2}O_{5}^{-}\right)}$	(23)
HV_2_O_5_^−^ + 3H_2_O = 2VO_4_^3−^ + 7H^+^ + 2*e*^−^	$E=1.962-0.2068\mathrm{pH}+0.0295\mathrm{lg}\frac{{\left({\mathrm{VO}}_{3}^{3-}\right)}^{2}}{\left({\mathrm{HV}}_{2}O_{5}^{-}\right)}$	(24)

**Table 2 molecules-28-02706-t002:** Main electrochemical reaction equations of Al-H_2_O system.

Electrochemical Reaction Equation	Eh-pH	Number
Al^3+^ + 3*e*^−^ = Al	E = −1.688 + 0.0197lg (Al^3+^)	(25)
Al(OH)_3_ + 3H^+^ = Al^3+^ + 3H_2_O	pH = 2.64 − 1/3lg (Al^3+^)	(26)
Al(OH)_3_ + 3H^+^ + 3*e*^−^ = Al + 3H_2_O	E = −1.532 − 0.0591pH	(27)
Al(OH)_3_ + 3H_2_O = Al(OH)_4_^−^ + 4H^+^	pH = 12.239 + 1/4lg (Al(OH)_4_^-^)	(28)

**Table 3 molecules-28-02706-t003:** Main electrochemical reaction equations of Cr-H_2_O system.

Electrochemical Reaction Equation	Eh-pH	Number
O_2_ + 4H^+^ + 4*e*^−^ = H_2_O	E = 1.228 − 0.0591pH	(44)
2H^+^ + 2*e*^−^ = H_2_	E = −0.0591pH	(45)
Cr^2+^ + 2*e*^−^ = Cr(s)	E = −0.913 + 0.0296lg (Cr^2+^)	(46)
Cr^3+^ + *e*^−^ = Cr^2+^	E = −0.42 + 0.0591 lg ($\frac{{\mathrm{Cr}}^{3+}}{{\mathrm{Cr}}^{2+}}$)	(47)
H_2_CrO_4_ + 6H^+^ + 3*e*^−^ = Cr^3+^ + 4H_2_O	E = 1.32 − 0.0197lg (Cr^3+^) − 0.1182pH	(48)
HCrO_4_^−^ + 7H^+^ + 3*e*^−^ = Cr^3+^ + 4H_2_O	E = 1.35 + 0.0197lg ($\frac{{\mathrm{HCrO}}_{4}{}^{-}}{{\mathrm{Cr}}^{3+}}$) − 0.1379pH	(49)
CrO_4_^2−^ + 4H^+^ + 3*e*^−^ = Cr(OH)_4_^−^	E = 0.915 + 0.0197lg ($\frac{{\mathrm{CrO}}_{4}{}^{2-}}{\mathrm{Cr}{\left(\mathrm{OH}\right)}_{4}{}^{-}}$) − 0.0788pH	(50)
H_2_CrO_4_ = HCrO_4_^−^ + H^+^	pH = 0.46 − lg (HCrO_4_^−^)	(51)
HCrO_4_^−^ = CrO_4_^2−^ + H^+^	pH = 6.52 + lg ($\frac{{\mathrm{HCrO}}_{4}{}^{-}}{{\mathrm{CrO}}_{4}{}^{+}}$)	(52)

**Table 4 molecules-28-02706-t004:** Main chemical multi-element analysis of original samples/%.

Component	V_2_O_5_	SiO_2_	Al_2_O_3_	Fe	MgO	CaO
Content	1.96	86.49	3.78	2.43	0.51	1.44
**Component**	**P_2_O_5_**	**S**	**Cr**	**K_2_O**	**Fcad**	**TiO_2_**
Content	0.68	0.13	0.062	0.56	0.90	0.034

**Table 5 molecules-28-02706-t005:** Valence distribution of original samples.

Project	Valence State			Total
	V(III)	V(IV)	V(V)	
Content/%	0.82	0.25	0.030	1.1 (Equivalent V_2_O_5_ is 1.96)
Distribution rate/%	74.55	22.73	2.72	100

**Table 6 molecules-28-02706-t006:** Vanadium phase analysis.

Mineral Phases	Vanadium-Bearing Muscovite	Garnet	Vanadium-Bearing Biotite	Vanadium-Bearing Iron Oxide	Vanadium-Bearing Chlorite
Distribution rate/%	49.36	21.10	17.26	3.81	3.03
Mineral phases	Vanadium-bearing ilmenite	Vanadium-bearing wollastonite	V_2_O_5_	Other	Total
Distribution rate/%	0.18	0.81	0.49	3.96	100.00

## Data Availability

Data will be made available on request.

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
