# Peer review of "Vanadium Extraction from the Flotation Concentrate of Vanadium-Bearing Shale by Process of Non-Roasting Enhanced Acid Leaching and Thermodynamics"

_molecules, 2023, doi:10.3390/molecules28062706_

Round 1
Reviewer 1 Report
I consider that the document is very long and can be presented in a shorter form without fundamentally altering its content. In my opinion, many of the reactions and tables can be omitted by simply referencing them. As it is presented, much relevance is given to the theoretical content of the thermodynamic process and it is preferred over the experimental results. I consider that this could be changed and in that way the work could be read more easily.
The conclusions should refer to the objectives that were initially proposed. Also the conclusions should focus on the experimental aspects and less on the theoretical ones.
From the point of view of form, it should be made clear that tables and figures do not show things. It should be said that the tables and figures do show things.
Author Response
Response to Reviewer 1 Comments
Point: I consider that the document is very long and can be presented in a shorter form without fundamentally altering its content. In my opinion, many of the reactions and tables can be omitted by simply referencing them. As it is presented, much relevance is given to the theoretical content of the thermodynamic process and it is preferred over the experimental results. I consider that this could be changed and in that way the work could be read more easily.
The conclusions should refer to the objectives that were initially proposed. Also the conclusions should focus on the experimental aspects and less on the theoretical ones.
From the point of view of form, it should be made clear that tables and figures do not show things. It should be said that the tables and figures do show things.
Response:
We are glad that the reviewer can make a professional evaluation. References are added to the thermodynamic theory part of the manuscript. The theory of the manuscript is to support the feasibility of the experiment and reveal the mechanism of the whole experiment. Therefore, there are many theoretical aspects.
Reviewer 2 Report
On page 3 (lines 109-110), the authors claim “Among them, garnet is an island aluminosilicate minerals, which belongs to insoluble mineral.” The garnets can be dissolved (https://www.eichrom.com/wp-content/uploads/2018/02/CJ106-Connelly-Gnt-Lu-Hf-compiled.pdf ). Please specify to what reagents (acids) is not soluble the garnet. The authors should also discuss the garnet dissolution and put it in perspective.
The Experimental section is missing important information about the origin of the reagents, concentrations, etc. It is also missing parameters of data collection. For example, what are the step size, time per step, and 2-theta range for the collection of the XRD? The authors should duly describe their experiments.
Fig. 7. In the XRD pattern, there are unidentified peaks. Why?
On page 20. I wonder how the SEM analysis can “confirm that F- can break the Si-O and Al-O bonds in vanadium-bearing mica and help H+ to strengthen”. Please re-write the sentence.
The SEM images are not convincing. Please show a more general image to support the description of fig. 10.
Fig.7. The authors should indicate the PDF cards used for phase identification.
The authors claim “XRD analysis showed that the leaching aids addition in acid leaching can weaken the intensity of the characteristic diffraction peaks of quartz, illite, and other minerals in samples.”. The change in the peak intensity should be quantified. How much decrease and what about other reasons?
The interpretation of the FTIR spectra is not convincing. FTIR gives information about the local structure, not the crystal structure (long-range atomic order).
Author Response
Response to Reviewer 2 Comments
Point 1: On page 3 (lines 109-110), the authors claim “Among them, garnet is an island aluminosilicate minerals, which belongs to insoluble mineral.” The garnets can be dissolved (https://www.eichrom.com/wp-content/uploads/2018/02/CJ106-Connelly-Gnt-Lu-Hf-compiled.pdf ). Please specify to what reagents (acids) is not soluble the garnet. The authors should also discuss the garnet dissolution and put it in perspective.
Response 1:
We are glad that you have carefully reviewed the manuscript. This sentence is a translation error. Garnet is not an insoluble species, but is difficult to dissolve in water and acid solution. This issue has been changed and marked in the manuscript.
Point 2: The Experimental section is missing important information about the origin of the reagents, concentrations, etc. It is also missing parameters of data collection. For example, what are the step size, time per step, and 2-theta range for the collection of the XRD? The authors should duly describe their experiments.
Response 2:
We appreciate your professional review of the manuscript. We have added relevant information and marked it in the manuscript.
Point 3: On page 20. I wonder how the SEM analysis can “confirm that F- can break the Si-O and Al-O bonds in vanadium-bearing mica and help H+ to strengthen”. Please re-write the sentence.
Response 3:
We are very happy to have your professional comments on our manuscript. The conclusion that fluorine ions can destroy Si-O and Al-O bonds is inferred from the combination of thermodynamic calculation results and SEM results. We restated this sentence.
Point 4: The SEM images are not convincing. Please show a more general image to support the description of fig. 10.
Response 4:
For your valuable comments, we have replaced the pictures.
Point 5: The SEM images are not convincing. Please show a more general image to support the description of fig. 10.
Response 5:
We thank you for pointing out the problem. The SEM image has been replaced in the manuscript.
Point 6: Fig.7. The authors should indicate the PDF cards used for phase identification.
Response 6:
We thank you for your valuable comments. We have added PDF card information to the manuscript.
Point 7: The authors claim “XRD analysis showed that the leaching aids addition in acid leaching can weaken the intensity of the characteristic diffraction peaks of quartz, illite, and other minerals in samples.”. The change in the peak intensity should be quantified. How much decrease and what about other reasons? The interpretation of the FTIR spectra is not convincing. FTIR gives information about the local structure, not the crystal structure (long-range atomic order).
Response 7:
Thank you for your professional comments. We have made changes and marked them.
Reviewer 3 Report
See file "molecules-2218532 remarks.docx"

Author Response
Response to Reviewer 3 Comments
Point 1: Line 60: “… shows stronger oxidation …”. Stronger than what? CaF2 is not an oxidation agent. Table 2: How was the valence distribution measured or determined?
Response 1:
We appreciate your valuable comments. We have adjusted the statement in the introduction and corrected the mistakes in word order. In addition, the distribution of valence states and the measured valence states in Table 2 are characterized by a special vanadium ore detection institute.
Point 2: Chapter 3.1.1: The rule that a reaction proceeds spontaneously if its Gibbs energy is below zero (△G < 0) is only valid if all species (reactants and products) exists in pure form exhibiting activities of unity (ai = 1). Otherwise, the law of mass action and the connection between G and K, the equilibrium constant, has to be considered. Furthermore, reaction 2 and also reaction 4 interfuse something. Correct is for the enthalpy as a function of temperature and for the enthalpy of a reaction If combined, the difference between products and reactants has to be calculated not only for the temperature dependent term (integral), but also for the term at standard conditions (298 K).
Response 2:
We appreciate your careful review of the manuscript. We have calculated the thermodynamic parameters at 298 K. For the calculation of △ G, 298 K is not included. This is to consider the range of temperature in the test, and 298k is not selected. However, from the calculation results, we can see the trend of several straight lines in the figure, and we can judge that 298 K is absolutely possible, which is consistent with common sense.
Point 3: Table 4 and Figure 1: The table can be deleted because the same information is already given by the EpH-diagram and the thermodynamic data are much too inaccurate to give exact values for the individual equilibria. The solid vanadium oxides (VO, V2O3, V2O4 and V2O5) are missing as well as the total V concentration, the molality. These vanadium oxides are hard to almost insoluble in water and consequently will appear in approximately neutral solutions. Furthermore, no information is given about the total V concentration, for which the diagram is calculated, although some of the presented reaction equilibria depend on it.
A similar situation applies for Table 5 and Figure 2 as well as Table 6 and Figure 4.
Response 3:
We are very grateful for your professional comments. For the concentration of V and other issues, we have revised and added them in the manuscript.
Point 4: Reactions 29-43 and Fig. 3, etc.: It makes no sense to describe such complex systems (K-Al-Si-O-H only for muscovite, which has extended by Mn, Ca, S and F due to the applied additives and acids). For instance, figure 3 indicates that the reaction 39 to 43 are more and more thermodynamically favored. However, they also require more and more F-(a) as a reactant and so which reaction of these is dominant depends not only on its Gibbs function but also on the amount of available fluoride ions in the system. Furthermore, the reaction equations consider hydrofluoric acid as a reagent, but chapter 2.3 indicates that sulfuric acid was applied as a leaching agent. However, the presence of fluoride ions can only results from the leaching aid CaF2, but calcium (as well as manganese from the other leaching aid) is completely ignored by the reaction equations.
Response 4:
Thank you for your careful review of our manuscript. We accept your proposal, and we have made the following explanation for this problem: the leaching agent of this test is sulfuric acid, and HF in the thermodynamic calculation is generated by the combination of sulfuric acid and fluorine ion, and hydrofluoric acid is highly corrosive, which also plays a role in the structural destruction of minerals. The thermodynamic parameters of Ca ion are not calculated in this manuscript because its dissolution in solution is relatively simple. Therefore, the manuscript only describes the relevant thermodynamic parameters of the most critical ions.
Point 5: Chapter 3.1: The EpH diagram of the whole system is not a superposition of the EpH diagrams for the individual metals, such as V, Al and Cr. The reason therefore is that the diagrams for the individual metals did not consider any species (solid or aqueous ion), which contain more than one metal, such for instance Al2SiO5, etc. This additionally ignores completely the mutual influence of different metals onto each other in the solution. As a consequence, the dissolution behavior of the complex minerals, such as muscovite and garnet, cannot be derived from the applications of EpH diagrams for individual subsystems, for instance V-H2O, Al-H2O, etc.
General: Due to the limited temperature, hydrometallurgical processes such as leaching are much more dominated by kinetics than by thermodynamic equilibrium. Furthermore, the thermodynamic data for many species in aqueous solutions are comparable imprecise and partly not even all species are already identified. The measurement of such data is extremely challenging due the extreme slow equilibrium setting, etc. As a consequence, it makes little sense to focus on a thermodynamic description for such a complex system.
Response 5:
We appreciate your valuable comments on the manuscript. In response to this problem, we made the following response: because vanadium-bearing shale is a complex mineral, we know from the test results that it is composed of multiple components. Therefore, for the study of thermodynamics, we have adopted the corresponding pure minerals of each component for calculation, because the current database of thermodynamics calculation is relatively simple, and there are few relevant calculations for complex minerals. Therefore, we should pay attention to the research in this field to provide convenience for future thermodynamic calculation.
Round 2
Reviewer 2 Report
I do not agree with the claim on page 22 "FT-IR spectrum analysis confirmed that the structure of vanadium-bearing mica can be destroyed...."
The structural integrity should be confirmed by XRD, not by FTIR spectroscopy. Please confirm the claimed results by powder XRD analysis.
Furthermore, the SEM shows images and from these images can't be deduced (page 22) " ...break the Si-O and Al-O bonds..." . Please make corrections.
Author Response
Response to Reviewer Comments
Point 1: I do not agree with the claim on page 22 "FT-IR spectrum analysis confirmed that the structure of vanadium-bearing mica can be destroyed...."
The structural integrity should be confirmed by XRD, not by FTIR spectroscopy. Please confirm the claimed results by powder XRD analysis.
Furthermore, the SEM shows images and from these images can't be deduced (page 22) " ...break the Si-O and Al-O bonds..." . Please make corrections.
Response 1: Please provide your response for Point 1.
Thank you very much for your professional review. In response to your comments, we have revisited them in the manuscript, made corresponding modifications and marked them.
Reviewer 3 Report
Again, a complex system, where many different reactions can occur, cannot be described by individual reactions. The comparison of the Gibbs energy for different reactions gives not the information, which reaction will proceed and which not because the reactions will influence each other. As a very simple example: CH4(g) + O2(g) = CO2(g) + 2 H2(g), DG(298 K) = -343.64 kJ/mol and CH4(g) + O2(g) = C(s) + 2 H2O(g), DG(298 K) = -423.55 kJ/mol does not imply that methane will react with a limited amount of oxygen only to carbon and water vapour, because one has also to consider the equilibrium of C(s) + H2O(g) = CO(g) + H2(g), etc.
There is no answer, how the valence distribution was measured or determined.
The equation (2) is still confused.
The direction of a reaction depends only on whether DG is larger or lower than zero, if all species exhibit an activity of unity.
The solid oxides are still missing in the EpH diagrams for vanadium and chromium.
Author Response
Response to Reviewer Comments
Point 1: Again, a complex system, where many different reactions can occur, cannot be described by individual reactions. The comparison of the Gibbs energy for different reactions gives not the information, which reaction will proceed and which not because the reactions will influence each other. As a very simple example: CH4(g) + O2(g) = CO2(g) + 2 H2(g), DG(298 K) = -343.64 kJ/mol and CH4(g) + O2(g) = C(s) + 2 H2O(g), DG(298 K) = -423.55 kJ/mol does not imply that methane will react with a limited amount of oxygen only to carbon and water vapour, because one has also to consider the equilibrium of C(s) + H2O(g) = CO(g) + H2(g), etc.
Response 1: We thank you for your valuable comments on the manuscript. In response to this problem, we have made the following response: for the complex minerals such as vanadium-bearing shale, the corresponding pure minerals of each component have been calculated, because the current database of thermodynamic calculation is relatively simple, and there are few related calculations for complex minerals. The interaction between reactions has not been studied, and can be further explored in the future. Moreover, the influence of metal ions (Mn) on it has been reported in relevant literature. What we can know is that the increase of temperature is not conducive to the enhanced leaching of vanadium by leaching aids. Therefore, in the future, we will mainly consider that the enhanced acid leaching of vanadium-bearing shale is mainly controlled by kinetic factors.
Point 2: There is no answer, how the valence distribution was measured or determined.
Response 2: The content of Table 2 is that the content of V (III), V (IV) and V (V) in the sample of vanadium bearing shale flotation concentrate was determined by ammonium ferrous sulfate potentiometric titration.
Point 3: The equation (2) is still confused.
Response 3: Modifications have been made to Equation 2.
Point 4: The direction of a reaction depends only on whether DG is larger or lower than zero, if all species exhibit an activity of unity.
Response 4: Modifications have been made to Equation 2.
First of all, thank you very much for your review. In response to this opinion, we make the following explanation: the reactions involved in the manuscript are inferred, and we try to be as comprehensive as possible. Some species that can't occur (△ G>0) are not listed. What we compare is the reaction between the enhanced and non-enhanced acid leaching systems. It can be seen that the activity of the enhanced acid leaching system is higher than that of the non-enhanced acid leaching system (the △ G of the enhanced system is very small).
Point 5: The solid oxides are still missing in the EpH diagrams for vanadium and chromium.
Response 5: Thank you very much for your professional review. For this opinion, we make the following explanation: when we do thermodynamic calculations, we also consider that many species will appear in the Eh-pH system of V and Cr. At that time, we judged that there were multiple species. When we put them into the software for calculation, these species did not exist under specific conditions. Therefore, we have not included relevant chemical reactions in the manuscript. It is undeniable that the solid oxides you mentioned may exist, but they do not appear under the conditions (temperature, pressure, etc.) specified in the manuscript. The result of thermodynamic calculation is mainly auxiliary. In future work, we intend to analyze the leaching process in hydrometallurgy from the perspective of kinetics.